# Serum uric acid a depression biomarker

Xiandong Meng[1◉], Xia Huang[1◉], Wei Deng[1], Jiping Li[2]*, Tao Li[1]

1 Mental Health Center of West China Hospital, Sichuan University, Chengdu, Sichuan Province, China,
2 Nursing department of West China Hospital, Sichuan University, Chengdu, Sichuan Province, China

◉ These authors contributed equally to this work.
* jp-li@163.com

**Data Availability Statement:** All relevant data are within the paper and its Supporting Information files.

**Funding:** Financial support for this research was from Ministry of Science and Technology of People's Republic of China (National key research and development plan). Name of the project is

## Abstract

### Objective

We aimed to investigate the difference in serum uric acid(SUA)levels between subtypes of depression and normal population, and whether SUA can be used to identify bipolar disorder depressive episode and major depressive disorder and predict the length of hospital stay.

### Methods

1543 depression patients and 1515 healthy controls were obtained according to the entry and exclusion criteria from one mental health center of a tertiary hospital in southwestern China. The diagnosis and classification of depression was in accordance with ICD-10. The SUA value was derived from fasting plasma samples analysis. The level of SUA of all the participants was quantified using Roche cobas8000-c702-MSB automatic biochemical analyzer. Data were analyzed by SPSS18.0 statistical software package.

### Results

Overall, the level of SUA in patients with depression was lower than that in normal control. Specifically, males' SUA levels were in the interval of [240, 323.3) and [323.3, 406.6), and women were in the [160, 233.3] levels. The SUA level of bipolar disorder depressive episode was higher compared to major depressive disorder level. Interestingly, male patients who were hospitalized for two weeks had higher SUA than those who were hospitalized for three weeks or four weeks.

### Conclusions

Our results suggest that the length of hospital stay may be associated with SUA, and when it is difficult to make a differential diagnosis of bipolar disorder depressive episode and major depressive disorder, the level of SUA may be considered. The adjustment of SUA as a method for treating depression needs to be carefully assessed.

study on the precise diagnosis and treatment model of schizophrenia based on multi-omics map, and the item number is 2016YFC0904300. The funders had no role in study design, data collection and analysis, decision to publish, or preparation of the manuscript.

**Competing interests:** The authors have declared that no competing interests exist.

# Introduction

Depression has a high prevalence and is a heavy disease burden. According to statistics, its global prevalence rate is about 4% to 5% [1], and about 10% to 20% of people have had various types of depression in their lifetime [2,3]. It is predicted that by 2020 [4]depression will rank second in the global burden of disease. Depression is thought to be the result of genetic predisposition combined with environmental interactions, and the oxidative stress may be one of its pathogenesis [5,6]. Oxidative stress can lead to decreased brain neurogenesis and increased neuronal apoptosis [7], and it can affect the activity of 5-HT neurotransmitters and the metabolic pathways of monoamine neurotransmitters [8]. In patients with depression, oxidative stress and lipid per oxidation are enhanced and the total antioxidant capacity is reduced [5,6], which in turn leads to various depressive symptoms in patients.

Uric acid is the end product of purine metabolism, which has endogenous or exogenous origin. Endogenous uric acid is produced primarily by the degradation of nucleic acids, adenine, and guanine by normal or impending dying cells. Exogenous uric acid is mainly produced by the synthesis of purine produced by animal proteins in the liver, small intestine, muscle tissue, kidney and vascular endothelium [9,10]. Uric acid is mainly excreted through the kidneys. Increased Serum uric acid (SUA) levels may cause gout, dyslipidemia, cardiovascular disease, high blood pressure, and other physical diseases [11–13]. On the other hand, decreased SUA levels may cause neurodegenerative diseases such as Alzheimer's disease, Huntington's disease, Parkinson's disease, and multiple sclerosis [14–16]. SUA is a strong antioxidant that provides more than 60% antioxidant activity in plasma [17,18]. The major mechanisms of uric acid function have been reported including: maintenance of peroxidase activity, which in turn prevents the formation of superoxide and peroxynitrite [19]; prevention of cellular enzymes in activation caused by peroxynitrite, which can result in damage to the cytoskeleton [20]; and inhibition of iron-dependent ascorbate oxidase by binding to iron, thereby preventing damage caused by oxidative stress [21].

There is no consistent pattern for SUA levels in patients with depression. Studies have shown that SUA in depression can be excessively consumed due to an increase in the anti-oxidative stress, and showed a downward trend [22], however, other studies have found that SUA levels in patients was higher than controls [23]. The reasons for this discrepancy can be associated with several factors; First, in published studies, different diagnostic criteria for depression was used. Currently there are 2widely accepted diagnostic criteria for depression: the International Classification of Diseases-10 (ICD-10) of Mental and Behavioural Disorders, and the Diagnostic and Statistical Manual of Mental Disorders-V (DSM-V). Importantly, both classifications differ greatly in the diagnostic criteria for depression [24]. In addition, depression has a variety of clinical subtypes and SUA may vary in the different subtypes, but previous studies have not considered these parameters. The present study includes patients diagnosed with depression according to ICD-10 diagnostic criteria when they are discharged; it aims to explore the difference of SUA levels between normal control and patients with depression of different subtypes. The feasibility of using SUA as a biomarker of depression is discussed.

# Materials and methods

## Study design and population

Patient samples were obtained from the Depression Database in the Research Data Management Platform of Mental Health Center of West China Hospital of Sichuan University. The database recorded all the patients of the Mental Health Center of West China Hospital of Sichuan University since 2009. In this study, inpatients diagnosed for recurrent depressive episode,

major depressive disorder, depression with anxiety, bipolar disorder depressive episode according to ICD-10between 2015.1.1–2017.12.31 were included. The healthy control samples were derived from the Laboratory Medicine Center of the West China Hospital of Sichuan University between 2017.1.1–2017.12.31. Exclusion criteria are: repeated hospitalization or multiple physical examinations, age less than 18, lack of SUA data, kidney disease (chronic renal failure, nephritis, nephrotic syndrome, etc.), history of taking drugs that can affect SUA, allergic diseases, metabolic diseases (gout, hyperuricemia), pregnancy or lactation and past smoking history. Overall, samples from a total of 1543 patients and 1515healthy control were obtained.

Patients were treated with antidepressants or mood stabilizers. Most of the antidepressants used were SSRIs such ascitalopram, escitalopram, paroxetine, sertraline. Others including tricyclic antidepressants, monoamine oxidase inhibitors, and other antidepressants such as Doxepin, Morclobemide, Trazodone and Agomelatine. The most commonly used mood stabilizers were lithium carbonate, sodium valproate and carbamazepine.

### Data collections and measures

Demographic characteristics included: gender, age, marriage, ethnicity, education level, length of hospital stay, all data were directly retrieved from the data management platform. SUA data was derived from biochemical examination of blood samples, fasting and empty stomach the day before the blood draw. The specimens were obtained by the nurses of the ward or the medical examination center. The drawing time was about 7:30 in the morning. The blood collection process was completed following a standardized blood specimen collection process developed by the Laboratory Medicine Center of West China Hospital of Sichuan University certified by CPA International. The collected blood samples were sent to the center by the staff of the transfer department within half an hour after the blood collection. The specimens were analyzed by the Roche cobas8000-c702-MSB automatic biochemical analyzer detection system on the same day. The normal values of SUA are considered to be240-490μmol/Land 160–380μmol/L for male and female respectively.

The study was approved by the Medical Ethics Committee of West China Hospital of Sichuan University and implemented in accordance with relevant ethical requirements. The authors had access to information that could identify individual participants during or after data collection.

### Statistical analysis

The demographic characteristics of the patient group and the normal control group were compared by chi-square test, and the SUA level was compared by independent sample t test. The comparison between the subtype of depression and the normal control group was assessed by analysis of variance and further by LSD (Least Significant Difference) test. Differences in the SUA levels between patients with different hospital length of stays and normal controls were analyzed by analysis of variance and further by LSD test. All of the above analyses used the SPSS18.0 statistical package. The criterion for statistical significance was $p < 0.05$.

## Results and discussion

### Baseline characteristics

Patient's and the normal population's age concentrated at 18–34 years, The distribution of other age groups in the two groups was basically the same (Table 1). Most of the participants got married and the percentage was 67.1% and 70.3% in normal population and patient

**Table 1. Comparison of demographic characteristics between patient population and normal population.**

| item | categories | normal | | patient | | $\chi^2$ | P |
|---|---|---|---|---|---|---|---|
| | | n | % | n | % | | |
| age | 18–34 | 458 | 30.2 | 444 | 28.8 | 3.156 | 0.532 |
| | 35–44 | 222 | 14.7 | 249 | 16.1 | | |
| | 45–54 | 281 | 18.5 | 303 | 19.6 | | |
| | 55–64 | 277 | 18.3 | 259 | 16.8 | | |
| | ≥65 | 277 | 18.3 | 288 | 18.7 | | |
| gender | male | 800 | 52.8 | 468 | 30.3 | 159.087 | <0.001 |
| | female | 715 | 47.2 | 1075 | 69.7 | | |
| marriage | unmarried | 339 | 22.4 | 300 | 19.4 | 5.519 | 0.137 |
| | married | 1017 | 67.1 | 1084 | 70.3 | | |
| | divorced | 86 | 5.7 | 76 | 4.9 | | |
| | widowed | 73 | 4.8 | 83 | 5.4 | | |

population respectively(Table 1). Comparison of demographic characteristics between patient and normal population indicates no significant statistical difference in age(p = 0.137) and marriage (p = 0.532) (Table 1). The majority of the normal population was male (52.8%), while the majority of the patient population was female(69.7%), which was related to women's increased vulnerability to depression. The difference between the two groups was statistically significant (p<0.001)(Table 1).

## SUA levels between patients and control groups

Comparison of SUA level in the patient and control groups indicates a statistically significant lower levels in the patient group (p<0.001) (Table 2). In view of the large range of SUA reference values, it was necessary to further explore the exact range of SUA difference between the patient group and the normal control. For this purpose, the value of SUA was divided into five intervals as follow: the normal value of uric acid was equally divided into three intervals including [240–323.3), [323.3–406.6) and [406.6–490), below the lower limit of normal(<240) and, above the upper limit (>490) of normal. Comparison with normal controls was analyzed separately for male and female patients (Tables 3 and 4). SUA levels in the male patients and normal population was statistically significant in the [240–323.3) and [323.3–406.6) ranges (p<0.001) however the difference for the female patient and the normal population was only significant in the [160, 233.3] range (p<0.001).

**Table 2. Comparison of SUA value between patients and normal.**

| | overall | | male | | female | |
|---|---|---|---|---|---|---|
| | normal | patient | normal | patient | normal | patient |
| N | 1515 | 1543 | 800 | 468 | 715 | 1075 |
| *Mean* | 337.47 | 298.54 | 382.81 | 355.24 | 286.73 | 273.86 |
| *Sd* | 86.02 | 89.50 | 78.20 | 94.71 | 62.97 | 74.75 |
| *t* | 12.260 | | 5.326 | | 3.929 | |
| *p* | <0.001 | | <0.001 | | <0.001 | |

**Table 3. Comparison of serum uric acid (SUA) in male between patients and normal.**

| | SUA<240μmol/L | | 240≤SUA<323.3 | | 323.3≤SUA<406.6 | | 406.6≤SUA≤490 | | SUA>490 | |
|---|---|---|---|---|---|---|---|---|---|---|
| | normal | patient | normal | patient | normal | patient | normal | patient | normal | patient |
| n | 14 | 40 | 171 | 132 | 342 | 179 | 205 | 79 | 67 | 36 |
| *Mean* | 214.64 | 206.63 | 292.06 | 283.97 | 368.06 | 361.63 | 441.75 | 437.50 | 542.97 | 561.86 |
| *Sd* | 21.47 | 27.70 | 22.41 | 23.30 | 24.77 | 22.47 | 22.81 | 22.20 | 49.08 | 91.19 |
| *t* | 0.982 | | 3.060 | | 2.993 | | 1.415 | | -1.156 | |
| *p* | 0.331 | | 0.002 | | 0.003 | | 0.158 | | 0.254 | |

## SUA between subtypes of depression and normal people

Statistical analysis showed that SUA levels were different between depression subtypes and normal population (p<0.001) (Table 5). Further LSD analysis found that the difference of SUA between the patients with recurrent depressive episode, major depressive disorder, depression with anxiety and normal control was statistically significant (p<0.001). The difference of SUA was not statistically significant between bipolar disorder patients with depressive episode and normal controls (p = 0.223). Interestingly the difference of SUA between bipolar disorder with depressive episode and depressive episode patients was statistically significant (p<0.001).

## SUA between patients with different length of hospitalization and normal people

Statistical analysis showed that there was a statistically significant difference in SUA levels between patients' different hospital stays and normal population (p<0.001) (Table 6). Furthermore, LSD test found that among male, the difference of SUA between two and three weeks (p = 0.023) and between two and four weeks (p = 0.005)of hospitalization were statistically significant. However the difference between three weeks and four weeks of hospitalization was not statistically significant (p = 0.424). Among female, there was no statistically significant difference in SUA between patients with the different length of hospital stays (p>0.05).

## Discussion

Epidemiological data suggest that depression has a higher prevalence and morbidity in women. Indeed, the incidence in female is about twice than that in men [25,26]. The present study sampling method is based on the established inclusion and exclusion criteria. A total of 1,543 patients composed of1075 females and 468 males were analyzed, this represents a ratio of 2.3(Table 1). Interestingly this corroborates with reported epidemiological data, indicating that our sample population was representative. Given the difference in the reference values of

**Table 4. Comparison of abnormal values of serum uric acid (SUA) in female between patients and normal.**

| | SUA<160μmol/L | | 160≤SUA<233.3 | | 233.3≤SUA<306.6 | | 306.6≤SUA≤380 | | SUA>380μmol/L | |
|---|---|---|---|---|---|---|---|---|---|---|
| | normal | patient | normal | patient | normal | patient | normal | patient | normal | patient |
| n | 6 | 39 | 127 | 294 | 347 | 432 | 186 | 222 | 49 | 88 |
| *Mean* | 141.83 | 141.01 | 212.46 | 204.80 | 269.386 | 267.018 | 336.108 | 338.305 | 432.43 | 434.41 |
| *Sd* | 12.11 | 14.62 | 17.68 | 18.95 | 20.1367 | 21.0381 | 18.9725 | 20.5153 | 57.65 | 59.06 |
| *t* | 0.132 | | 3.880 | | 1.592 | | -1.122 | | -0.191 | |
| *p* | 0.896 | | <0.001 | | 0.112 | | 0.262 | | 0.849 | |

**Table 5. Comparison of SUA between subtypes of depression and normal people.**

| Subtypes of depression | N | Mean | Sd | F | P |
|---|---|---|---|---|---|
| recurrent depressive episode | 220 | 287.01 | 83.68 | 44.448 | <0.001 |
| major depressive disorder | 1066 | 298.50 | 89.61 | | |
| depression with anxiety | 105 | 279.89 | 80.26 | | |
| bipolar disorder depressive episode | 152 | 328.40 | 96.05 | | |
| normal | 1515 | 337.47 | 86.02 | | |

Further LSD test found:

The difference of SUA between the patients with recurrent depressive episode, major depressive disorder, depression with anxiety and normal was statistically significant (p<0.001).

The difference of SUA was not statistically significant between the patients with bipolar disorder depressive episode and normal (p = 0.223).

The difference of SUA between patients with bipolar disorder depressive episode and major depressive disorder was statistically significant (p<0.001).

SUA between female and male genders and the gender composition ratio within the patient groups, the study was stratified by gender, followed by analysis of SUA levels.

Oxidative stress is one of depression pathogenesis. Excessive oxidative stress leads to impaired brain function in patients, leading to various psychiatric symptoms. SUA is a strong antioxidant [27] and is constantly consumed to counterattack the oxidative stress. This study showed that SUA level in depression and normal control group was different and statistically significant (p<0.001), and that the level in patients with depression was lower than that in normal people (Table 2). This finding is consistent with previous research reports. It confirms the role of oxidative stress in the pathogenesis of depression, and also suggests that SUA levels can be used as a biomarker for depression. Because SUA is excessively consumed, the antioxidant function in the brain is reduced leading to the onset of the disease. Therefore, adjusting the SUA level in patients might restore the ability of antioxidant stress and protect the brain function in patients with depression. Given the wide range of normal SUA values, it is worth exploring further the utility of this marker. The present study analyzed the results based on 5 intervals ranges of normal SUA and according to genders. The statistical analysis showed that the difference in SUA levels between the male patient group and the normal control group was statistically significant in two 2 SUA level interval ranges: the [240, 323.3) and [323.3, 406.6) (p<0.001) (Table 3). Whereas the difference in SUA levels between the female patient group

**Table 6. Comparison of SUA between patients with different hospital stay\* and normal people.**

| | male | | | | female | | | |
|---|---|---|---|---|---|---|---|---|
| | two weeks | three weeks | four weeks | normal | two weeks | three weeks | four weeks | normal |
| N | 136 | 163 | 112 | 800 | 311 | 408 | 231 | 715 |
| Mean | 371.13 | 348.93 | 340.66 | 382.81 | 269.12 | 273.33 | 278.72 | 286.73 |
| Sd | 96.9 | 99.45 | 85.02 | 78.2 | 67.73 | 76.81 | 71.65 | 62.97 |
| F | 13.589 | | | | 6.098 | | | |
| P | <0.001 | | | | <0.001 | | | |

Further LSD test found:

Among male, the difference of SUA between the two weeks of hospitalization and the three weeks was statistically significant(p = 0.023). The difference of SUA between the two weeks of hospitalization and four weeks of hospitalization was also statistically significant (p = 0.005). The difference between the three weeks of hospitalization and the four weeks of hospitalization was not statistically significant (p = 0.424).

Among female, there was no statistically significant difference in SUA between patients with different hospital stays (p>0.05).

\*Patients hospitalized for one week or longer than four weeks were excluded from statistical analysis, cause of the too small sample size.

and the normal control group was statistically significant in the [160, 233.3) interval (p<0.001) (Table 4). These type of analysis and findings have not been mentioned in previous studies. This result indicates that SUA levels in patients with depression show differences compared to normal population only within a specific narrow range of SUA levels. This type of analysis should be taken in consideration when using SUA as a biomarker for depression, even though it was significant only in a small part of the patients. This finding also indicate the importance of identifying additional biological markers for depression.

In recent years, modulation of SUA level was thought to represent a novel target for affective disorder treatment [28,29]. Although elevated SUA levels may have a protective effect on the central nervous system and may reduce the incidence of depression, on another hand high SUA levels can lead to increased morbidity and mortality due to several pathologies including hypertension, gout, coronary heart disease, stroke and other diseases induced by excessive SUA level. Therefore, the ideal level of SUA should be able to procure central nervous system protection while reducing the risk of inducing other diseases [30]. If the regulation of SUA levels is used as a treatment for depression, then the possible effect of SUA on the other potential diseases should be taken into account and the benefit for patients should be weighed instead of blindly regulating this marker by diet or medication, The current antioxidant treatments, whether given in prophylactic or therapeutic setting, are not always beneficial and can be harmful for human health [31]. According to this study findings, if the adjustment of SUA level would be used as a treatment for depression, it is recommended to intervene only within patients having circulating SUA levels within the lower normal range.

As per ICD-10 diagnostic criteria, depression includes multiple subtypes, and the differences in SUA levels between the normal populations and the disease subtypes have been less studied in previous studies. The present study found that the difference between SUA in patients with recurrent depressive episode, major depressive disorder, depression with anxiety compared to normal control was statistically significant (p<0.001) (Table 5). This is consistent with previous reports, but the difference of SUA between patients with bipolar disorder depressive episode and normal controls was not statistically significant (p = 0.223) (Table 5). Previous studies showed that SUA production was increased in mania patients due to the dysfunction of purinergic system, and the SUA levels were higher than normal [32,33]. Patients with depression have an increased SUA consumption due to oxidative stress [22] which in turn shows lower levels of SUA compared to normal population [34,35]. There is no difference in SUA between bipolar disorder depressive episode and normal populations. One explanation for this later finding is that the production and the consumption of SUA might occur at the same time, in order to maintain a balance. Based on this, we could assume that when the production of SUA is higher than the consumption, the symptom of mania will display more, and when the production of SUA is lower than the consumption, it's the symptom of depression that will show more in bipolar disorder depressive episode patients. This advanced hypothesis requires further research.

The difference of SUA between patients with bipolar disorder depressive episode and major depressive disorder was statistically significant (p<0.001)(Table 5). According to this result, It could be hypothesized that SUA may be a biomarker for the differentiation of unipolar and bipolar affective disorder. In outpatients and inpatients, as well as in children and adolescents, bipolar disorder with depressive episodes is often misdiagnosed as major depressive episodes [36]. This is due to the doctor's personal interpretation of the diagnostic criteria and symptom description given the patients. In addition, the absence of biomarker for reference makes it difficult to make the right diagnosis which increases the ambiguity in making the right diagnosis. It is necessary to do more researches to further show the specific efficacy of SUA as a biomarker for distinguishing these two diseases in future.

Hospitalization is an important part for treatment of acute depression. The length of hospital stay is directly correlated to the severity of the illness. The results showed that among male, the difference of SUA between the two weeks of hospitalization and three weeks (p = 0.023) and between two and five weeks (p<0.001) was statistically significant (Table 5). In general, patients with mild illnesses have shorter hospital stays, and those with severe illnesses have longer hospital stays. The SUA level in male patients who were hospitalized for two weeks was higher than in those hospitalized for three or four weeks, indicating that SUA was more consumed as the condition worsened. The average hospitalization day is currently the focus of government, patients and medical institutions. How to better predict the hospitalization time of patients is a problem worth exploring, however it lacks objective monitoring parameters, according to the result of this study, SUA level may be considered.

This study presents several limitations. First, it's a single-center study, mainly involving Han nationality patients; however given that China is a multi-ethnic country with a large population, it's impossible to have a nation-wide representative population. Secondly, the current research included a single biomarker. For a more comprehensive analysis, identifications of additional biomarkers warrant consideration. Furthermore, this study used patient hospital discharge diagnosis to identify the target population; however the patient population was treated by different doctors within the same hospital. The diagnosis of depression and depression subtypes might be different between different doctors, and this might affect the overall research results. It will be interesting the run the same study with patients' samples from same center and treated with same doctor.

## Conclusions

Overall, our analysis showed that SUA level in patients with depression is lower than that in normal population, but this difference is mainly reflected in the lower level of normal SUA value. This finding should be considered by the medical staff treating patients. The length of hospital stay may be associated with SUA, and when it is difficult to make a differential diagnosis of bipolar disorder depressive episode and major depressive disorder, the level of SUA may be considered. Although we can't draw the conclusion that SUA is a biomarker of depression according to the present results, it has certain reference value for subsequent related researches.

## Supporting information

**S1 File. Data used in this study.**
(SAV)

## Author Contributions

**Conceptualization:** Xiandong Meng, Xia Huang, Wei Deng, Jiping Li, Tao Li.

**Data curation:** Xiandong Meng, Xia Huang.

**Formal analysis:** Xiandong Meng.

**Funding acquisition:** Tao Li.

**Investigation:** Xiandong Meng.

**Methodology:** Xiandong Meng, Xia Huang, Wei Deng.

**Project administration:** Xiandong Meng.

**Resources:** Xiandong Meng.

**Supervision:** Jiping Li.

**Writing – original draft:** Xiandong Meng.

**Writing – review & editing:** Jiping Li.

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
