## [Decision Letter · Decision Letter 0]

8 Jan 2020

PONE-D-19-30310

Serum uric acid a depression biomarker

PLOS ONE

Dear Mr. Meng,

Thank you for submitting your manuscript to PLOS ONE. After careful consideration, we feel that it has merit but does not fully meet PLOS ONE’s publication criteria as it currently stands. Therefore, we invite you to submit a revised version of the manuscript that addresses the points raised during the review process.

The reviewers addressed some major and minor concerns about your manuscript. Please revise your manuscript carefully.

We would appreciate receiving your revised manuscript by Feb 22 2020 11:59PM. To enhance the reproducibility of your results, we recommend that if applicable you deposit your laboratory protocols in protocols.io, where a protocol can be assigned its own identifier (DOI) such that it can be cited independently in the future. For instructions see: http://journals.plos.org/plosone/s/submission-guidelines#loc-laboratory-protocols

We look forward to receiving your revised manuscript.

Kind regards,

Kenji Hashimoto, PhD

Academic Editor

PLOS ONE

Journal Requirements:

2. Thank you for stating the following beneath the Acknowledgments Section of your manuscript:

'Funding

Financial support for this research was from Ministry of Science and Technology of People's

Republic of China (National key research and development plan). Name of the project is study on

the precise diagnosis and treatment model of schizophrenia based on multi-omics map, and the

item number is 2016YFC0904300.'

Please provide an amended Funding Statement that declares *all* the funding or sources of support received during this specific study (whether external or internal to your organization) as detailed online in our guide for authors at http://journals.plos.org/plosone/s/submit-now 

Please state what role the funders took in the study.  If any authors received a salary from any of your funders, please state which authors and which funder. If the funders had no role, please state: "The funders had no role in study design, data collection and analysis, decision to publish, or preparation of the manuscript."

4. We note you have included a table to which you do not refer in the text of your manuscript. Please ensure that you refer to Table 1 in your text; if accepted, production will need this reference to link the reader to the Table.

5. Please include your tables as part of your main manuscript and remove the individual files.

Please note that supplementary tables should be uploaded as separate "supporting information" files.

6. Please include captions for your Supporting Information files at the end of your manuscript, and update any in-text citations to match accordingly. Please see our Supporting Information guidelines for more information: http://journals.plos.org/plosone/s/supporting-information

Reviewers' comments:

Reviewer's Responses to Questions

**Comments to the Author**

1. Is the manuscript technically sound, and do the data support the conclusions?

Reviewer #1: No

Reviewer #2: Partly

Reviewer #3: Yes

2. Has the statistical analysis been performed appropriately and rigorously? 

Reviewer #1: Yes

Reviewer #2: No

Reviewer #3: Yes

3. Have the authors made all data underlying the findings in their manuscript fully available?

Reviewer #1: No

Reviewer #2: No

Reviewer #3: No

4. Is the manuscript presented in an intelligible fashion and written in standard English?

Reviewer #1: Yes

Reviewer #2: No

Reviewer #3: Yes

5. Review Comments to the Author

Reviewer #1: The authors examined the difference in serum uric acid levels between depression and normal population in a large cohort, and they demonstrated that the levels of serum uric acid was lower than that in normal control, as well as higher serum uric acid levels in bipolar depressive episode compared to major depressive disorder. Furthermore, they demonstrated higher serum uric acid levels in male patients who were hospitalized for two weeks than in those who were hospitalized for three weeks or four weeks.

This study is interesting, and this study includes positive findings.

I wrote several comments below.

1. In methods, they should write how to treat the blood in detail (drawing time, processing and storage conditions).

2. In methods, there are no information of patient`s medication, such as antidepressants or mood stabilizers).

3. In methods; how to evaluate non-psychiatric controls?

4. In methods; they wrote that they exclude repeated hospitalization. Did all patients who participated in this study admit first time? Please write this exactly.

5. In results; did raw serum uric concentration values did follow a Gaussian distribution? If not, the data should be natural log-transformed before the analysis.

6. In results and discussion; they often wrote that SUA can be used as a biomarker for depression. Although they showed that the levels of serum uric acid was lower than that in normal control, as well as higher serum uric acid levels in bipolar depressive episode compared to major depressive disorder, they did not show the usefulness of biomarkers. They should show a sensitivity and specificity.

7. In conclusion, they wrote that SUA can be biological marker to predict the severity of the disease. However, they did not show the severity of patients and treatments in this study. I think this conclusion is not proper.

8. In limitations, it is better to write that the differences in serum uric acid levels between bipolar depressive episode and major depressive disorder may reflect different treatments between them (not reflect different diagnosis).

Reviewer #2: The authors investigated the difference in serum uric acid concentration between subtypes of depression and normal population in a cohort comprising 1543 depression patients and 1515 healthy controls. They further analysed whether serum uric acid can be used to identify bipolar disorder depressive episode and major depressive disorder and also asked the question if it can be used as a biomarker to predict the length of hospital stay. They found lower concentrations of uric acid in patients with depression compared to the controls and higher concentrations in patients with bipolar disorder depressive episodes compared to patients with major depressive disorder.

The research question is interesting but I have several concerns:

- The differences in uric acid concentrations seem to be rather small even in those groups where the difference is statistically significant. The authors categorized the patients according to their uric acid concentration in five groups and then perform five t tests. I wonder if an adjustment for multiple testing would be appropriate here. Anyway, are the differences in uric acid concentrations clinically meaningful? How large is the overlap in the uric acid distributions between the cases and the controls?

- Was it really one of the aims to analyse uric acid as a biomarker for hospital stay? Or was this only added because the p value was significant? I would not present this result so prominently because I do not see the use of it.

- It would be a large improvement if the authors could replicate their findings in another study but I am aware of the fact that this might be difficult to accomplish.

- Patients with hyperuricemia have been excluded from the study. How many patients with depression also suffer from hyperuricemia?

- Do the authors have any information on other markers of oxidative stress?

- Only very few characteristics of the study population are presented and possible confounding cannot be assessed. Is information regarding inflammation markers of e.g. coronary heart disease available? Depression and coronary heart disease might be linked vi inflammation or one may increase the risk of developing the other.

- The authors first state that “all data are fully available without restriction” but then state that “Data cannot be shared publicly because of the request of our hospital”. So I think the first statement will have to be changed.

- There are some typos and repeatedly whitespaces are missing. The language should be revised. The unit of uric acid is given as umol//L instead of µmol/L.

- In all the tables as well as in the text a number of p values is reported as 0.00. I would recommend to report three digits. Please report the actual p value and use something like p<0.001 for very low p values.

- Table 1 is not referenced in the text.

- Why did the authors categorize age? They could use it as continuous variable.

- Was uric acid normally distributed? It would be interesting to look at the distribution in the different patient groups, perhaps in form of box plots.

Reviewer #3: The manuscript described a technically sound piece of scientific research with data that supports the conclusions. and the experiments have been conducted rigorously, with appropriate controls, replication, and sample sizes. The conclusions have drawn appropriately based on the data presented. The manuscript is technically sound, and the data support the conclusions. But the authors have not made all data underlying the findings in their manuscript fully available.

6. PLOS authors have the option to publish the peer review history of their article (what does this mean?). If published, this will include your full peer review and any attached files.

Reviewer #1: Yes: Shusuke Numata

Reviewer #2: No

Reviewer #3: No

---

## [Author Response · Author response to Decision Letter 0]

21 Jan 2020

PONE-D-19-30310

Serum uric acid a depression biomarker

PLOS ONE

Dear Mr. Meng,

Thank you for submitting your manuscript to PLOS ONE. After careful consideration, we feel that it has merit but does not fully meet PLOS ONE’s publication criteria as it currently stands. Therefore, we invite you to submit a revised version of the manuscript that addresses the points raised during the review process.

The reviewers addressed some major and minor concerns about your manuscript. Please revise your manuscript carefully.

We would appreciate receiving your revised manuscript by Feb 22 2020 11:59PM. To enhance the reproducibility of your results, we recommend that if applicable you deposit your laboratory protocols in protocols.io, where a protocol can be assigned its own identifier (DOI) such that it can be cited independently in the future. For instructions see: http://journals.plos.org/plosone/s/submission-guidelines#loc-laboratory-protocols

• A rebuttal letter that responds to each point raised by the academic editor and reviewer(s). This letter should be uploaded as separate file and labeled 'Response to Reviewers'.

• A marked-up copy of your manuscript that highlights changes made to the original version. This file should be uploaded as separate file and labeled 'Revised Manuscript with Track Changes'.

• An unmarked version of your revised paper without tracked changes. This file should be uploaded as separate file and labeled 'Manuscript'. 

We look forward to receiving your revised manuscript.

Kind regards,

Kenji Hashimoto, PhD

Academic Editor

PLOS ONE

Journal Requirements:

Reply

Amended. I downloaded the style templates from the website above and amended my manuscript (see pages in the Revised Manuscript with Track Changes)

2. Thank you for stating the following beneath the Acknowledgments Section of your manuscript Funding Financial support for this research was from Ministry of Science and Technology of People's Republic of China (National key research and development plan). Name of the project is study on the precise diagnosis and treatment model of schizophrenia based on multi-omics map, and the item number is 2016YFC0904300.

"The funders had no role in study design, data collection and analysis, decision to publish, or preparation of the manuscript.

a. Please provide an amended Funding Statement that declares *all* the funding or sources of support received during this specific study (whether external or internal to your organization) as detailed online in our guide for authors at http://journals.plos.org/plosone/s/submit-now

b. Please state what role the funders took in the study. If any authors received a salary from any of your funders, please state which authors and which funder. If the funders had no role, please state: "The funders had no role in study design, data collection and analysis, decision to publish, or preparation of the manuscript."

c. c. Please include your amended statements within your cover letter; we will change the online submission form on your behalf. 

Reply

Amended. I removed the funding-related text in the manuscript (see page10 in the Revised Manuscript with Track Changes) and uploaded the Funding Statement according to the guide for authors above. I included my Amended statements within my cover letter according to the suggestion from the editor (see Revised Cover letter).

3. We note that you have indicated that data from this study are available upon request. PLOS only allows data to be available upon request if there are legal or ethical restrictions on sharing data publicly. For information on unacceptable data access restrictions, please see 

http://journals.plos.org/plosone/s/data-availability#loc-unacceptable-data-access-restrictions.

Reply

Amended. At the time of submission, we did not fully understand the requirements for data upload. After consulting our hospital's management agency, the data used in this study can be uploaded. We have uploaded the data while uploading the revised manuscript and noted this in the cover letter.

4. We note you have included a table to which you do not refer in the text of your manuscript. Please ensure that you refer to Table 1 in your text; if accepted, production will need this reference to link the reader to the Table. 

Reply

Amended (see page5 in the Revised Manuscript with Track Changes)

5. Please include your tables as part of your main manuscript and remove the individual files. Please note that supplementary tables should be uploaded as separate "supporting information" files. 

Reply

Amended (see page5-6 in the Revised Manuscript with Track Changes) 

6. Please include captions for your Supporting Information files at the end of your manuscript, and update any in-text citations to match accordingly. Please see our Supporting Information guidelines for more information: http://journals.plos.org/plosone/s/supporting-information

Reply

Amended (see page12 in the Revised Manuscript with Track Changes)

Reviewers' comments:

Reviewer's Responses to Questions

Comments to the Author

1. Is the manuscript technically sound, and do the data support the conclusions?

Reviewer #1: No

Reviewer #2: Partly

Reviewer #3: Yes________________________________________

2. Has the statistical analysis been performed appropriately and rigorously? 

Reviewer #1: Yes

Reviewer #2: No

Reviewer #3: Yes 

3. Have the authors made all data underlying the findings in their manuscript fully available?

Reviewer #1: No

Reviewer #2: No

Reviewer #3: No

4. Is the manuscript presented in an intelligible fashion and written in standard English?

Reviewer #1: Yes

Reviewer #2: No

Reviewer #3: Yes 

5. Review Comments to the Author

Reviewer #1: 

The authors examined the difference in serum uric acid levels between depression and normal population in a large cohort, and they demonstrated that the levels of serum uric acid was lower than that in normal control, as well as higher serum uric acid levels in bipolar depressive episode compared to major depressive disorder. Furthermore, they demonstrated higher serum uric acid levels in male patients who were hospitalized for two weeks than in those who were hospitalized for three weeks or four weeks.

This study is interesting, and this study includes positive findings. I wrote several comments below.

1. In methods, they should write how to treat the blood in detail (drawing time, processing and storage conditions).

Reply

Amended (see page4 in the Revised Manuscript with Track Changes)

2. In methods, there are no information of patient`s medication, such as antidepressants or mood stabilizers).

Reply

Amended (see page3 in the Revised Manuscript with Track Changes)

3. In methods; how to evaluate non-psychiatric controls?

Reply

The non-psychiatric control group in this study was derived from the physical examination population in our hospital. Before the physical examination personal information registration was required, including personal past history, present history, and family history. In view of the possible stigma of patients with psychiatrics, there are two cases of the authenticity of the medical history information provided by the population. One is to provide a completely true history of the individual's psychiatric illness, and the other is to conceal the individual's history of the mental illness. When we designed the study, two different methods were used to solve it. For the former, they were excluded based on the records in the personal information registration form. For the latter, this study did not find a better solution except including them. The reasons are as follows: 1. Patient’s information sharing between medical institutions in China has not yet been realized, so it is impossible for us to know the patient's medical records in other medical institutions. 2. Illness is a patient's privacy. If the person seeking for medical examination is unwilling to provide it, the medical institution has no right to investigate further. But according to our experience we supposed that the person deliberately concealing their past and present history were very few, and for prudence, we consulted a statistician he supposed that the impact on the final study results was very small. 

4. In methods; they wrote that they exclude repeated hospitalization. Did all patients who participated in this study admit first time? Please write this exactly.

Reply

The inpatient information system used by our hospital can record the number of hospitalizations of each patient. According to this, for patients with only one record were selected, and for patients with several records only the relevant data for the first record were selected, but this did not guarantee that all patients included in this study would be their first hospitalization. The reasons are as follows: 1. The source of the data used in this research derived from the inpatient information system of our hospital. Before the system was used, our hospital used the paper version of the disease records. Therefore, the first hospitalized patient recorded in the information system may have been hospitalized in our hospital before. 2. China's hospitalization information for patients has not been shared between medical institutions. Patients may have been hospitalized in other hospitals before coming to our hospital for the first time. 

5. In results; did raw serum uric concentration values did follow a Gaussian distribution? If not, the data should be natural log-transformed before the analysis.

Reply

The histogram and P-P chart were used to judge the distribution of the uric acid value. According to the result, the values followed the Gaussian distribution.(Figure 1-2) 

Figure 1 P-P diagram and histogram of uric acid distribution in normal population

Figure 2 P-P diagram and histogram of uric acid distribution in patients

6. In results and discussion; they often wrote that SUA can be used as a biomarker for depression. Although they showed that the levels of serum uric acid was lower than that in normal control, as well as higher serum uric acid levels in bipolar depressive episode compared to major depressive disorder, they did not show the usefulness of biomarkers. They should show a sensitivity and specificity.

Reply

In results and discussion, the discussion of serum uric acid as a biological marker that distinguishes bipolar disorder depressive episode and major depressive disorder is indeed not deep enough. This is mainly related to the purpose of this study. Through the current design of this study, only preliminary discovery of serum uric acid may be a biological marker that distinguishes the two diseases. To assist clinical diagnosis, a large sample survey and relevant statistical analysis are needed to clarify the serum uric acid reference and the sensitivity and specificity of these two types of patients. After that the value of serum uric acid may help clinicians to better distinguish the two diseases based on the patient's symptoms and serum uric acid level. These will be the focus of our next research. 

7. In conclusion, they wrote that SUA can be biological marker to predict the severity of the disease. However, they did not show the severity of patients and treatments in this study. I think this conclusion is not proper.

Reply

The judgment of the severity of the patients in this study was based on the length of the patient's hospitalization. According to the principle of diagnosis and treatment, the clinician will continuously evaluate the patient's condition dynamically during the hospitalization. Therefore, the length of hospital stay can indirectly reflect the severity of the patient's condition. Generally speaking, patients with less severe illnesses have shorter hospital stays, and patients with more severe illnesses have longer hospital stays. Therefore, this study combined the differences in uric acid levels between patients with different lengths of hospital stay, and concluded that uric acid uric acid may be a biological indicator reflecting the severity of the patient's illness.

8. In limitations, it is better to write that the differences in serum uric acid levels between bipolar depressive episode and major depressive disorder may reflect different treatments between them (not reflect different diagnosis).

Reply

The uric acid data of patients in this study were obtained from blood specimens collected on the day of admission or the day after admission. At this time the patient did not receive treatment or the patient's medication has just begun. Therefore, we thought that the difference in uric acid levels between the bipolar disorder depressive episode and major depressive disorder at this time reflected the difference between the two diseases. Therefore, we have concluded that uric acid can be used as a reference for the differential diagnosis of these two diseases. 

Reviewer #2: 

The authors investigated the difference in serum uric acid concentration between subtypes of depression and normal population in a cohort comprising 1543 depression patients and 1515 healthy controls. They further analysed whether serum uric acid can be used to identify bipolar disorder depressive episode and major depressive disorder and also asked the question if it can be used as a biomarker to predict the length of hospital stay. They found lower concentrations of uric acid in patients with depression compared to the controls and higher concentrations in patients with bipolar disorder depressive episodes compared to patients with major depressive disorder.

The research question is interesting but I have several concerns:

- The differences in uric acid concentrations seem to be rather small even in those groups where the difference is statistically significant. The authors categorized the patients according to their uric acid concentration in five groups and then perform five t tests. I wonder if an adjustment for multiple testing would be appropriate here. Anyway, are the differences in uric acid concentrations clinically meaningful? How large is the overlap in the uric acid distributions between the cases and the controls?

Reply

From the overall comparison of the uric acid level between the normal and patient populations, the difference was statistically significant (see Table 2). Combined with previous research literature, the difference between the two groups, and clinician recommendations, we comprehensively consider the difference had clinical meaning, so the results of this statistical analysis were presented in the final paper. In view of the large range of uric acid reference values, previous studies have failed to determine the specific range of differences in uric acid values between the normal population and depression patients, so this study segmented them based on uric acid reference values and uric acid abnormality values, and finally the specific interval of the difference in uric acid value between the normal population and depression patients. Statistical analysis showed that after subdividing the uric acid value interval, the difference of the uric acid value between the normal population and the depression patient population was statistically significant. Although the value of the difference was small, the results of this study have explored the role of uric acid in the pathogenesis of depression. This may have certain clinical meaning.

- Was it really one of the aims to analyze uric acid as a biomarker for hospital stay? Or was this only added because the p value was significant? I would not present this result so prominently because I do not see the use of it.

Reply

The length of hospital stay is a focus of the society, the government, and medical institutions. Shortening the length of hospitalization is of great significance for reducing the burden of hospitalization for patients and the expenditure of government medical insurance. At present, medical staff's assessment of the expected length of stay of newly admitted patients is mainly based on personal experience and information obtained through interviews, physical examinations and medical history collection. This practice has certain shortcomings. If some kind of biological marker can be used as a reference, it is likely to improve the accuracy of prediction. So we considered it at the beginning of the study design. The reason for linking them was that, according to our clinical observation, the length of hospital stay of patients with severe depression was longer than that of patients with less severe illness. Based on the role of oxidative and antioxidant stress in the pathogenesis of depression, we speculated that patients with long hospital stays may have lower uric acid levels than those with short hospital stays. We therefore included the length of hospital stay and analyzed its relationship with uric acid. We only got the preliminary conclusions, which needed further verification in the future.

- It would be a large improvement if the authors could replicate their findings in another study but I am aware of the fact that this might be difficult to accomplish.

Reply

As pointed out by reviewing experts, there were some shortcomings in this study indeed, but after consulting clinicians and statisticians, we believed that these existing shortcomings would not have a fatal impact on the final research results. If possible, we will carry out one more rigorous designed study in conjunction with valuable suggestions from reviewers, and conduct a more comprehensive and in-depth verification of the research results.

- Patients with hyperuricemia have been excluded from the study. How many patients with depression also suffer from hyperuricemia?

Reply

The hyperuricemia in this study referred to patients who had been explicitly diagnosed with the disease in their past history at the time of admission. For those who had the first uric acid test result after hospitalization that was above the upper limit of normal value and reached the definition of uric acid level for hyperuricemia were included in this study and analyzed as a separate group (see Table 3-4). In this study, 26 patients were explicitly diagnosed to have hyperuricemia.

- Do the authors have any information on other markers of oxidative stress?

Reply

According to the results of the previous literature review and the supplementary review of the literature after obtaining this recommendation, the oxidative stress-related markers in patients with depression are arranged as follows:

(1) Malondialdehyde

MDA is the final decomposition product of the oxidation reaction of polyvalent unsaturated fatty acids in the body, which will cause cross-linking between DNA molecules, and also cause internal cross-linking of protein molecules to affect the quality and quantity of enzyme proteins, leading to changes in biofilm structure and increased permeability causing cell damage. Its content reflects the level of free radicals in the tissue and the degree of lipid peroxidation [1]. Significant increases in plasma malondialdehyde levels have been reported in patients with depression [2,3].

(2) Inflammatory cytokines

Studies have shown that although there is no widespread immune activation in patients with depression, but with the clinical symptoms of depression, the levels of pro-inflammatory cytokines such as IL-1β, IL-6 and TNFα increase in the blood [4]. After antidepressant treatment, the above indicators showed a downward trend, suggesting that these inflammatory factors may be indicators of the state of depression [5].

(3) Cyclooxygenase

Cyclooxygenase mainly catalyzes arachidonic acid to produce prostaglandins. Under stress, the expression of cyclooxygenase is significantly enhanced and it is involved in the regulation of inflammatory responses in many tissues. These reactions involve the occurrence and development of neuropsychiatric disorders such as depression and Alzheimer's disease [6,7].

(4) Active nitrogen

Nitric oxide has a neurotransmitter or modulatory effect. It transmits information between cells and regulates many behavioral and endocrine responses. But too much nitric oxide release is neurotoxic, leading to hippocampus damage and may play an important role in the pathogenesis of depression [8].

(5) Oxygen free radical

Excessive oxygen free radical can damage mitochondrial respiratory chain activity and mitochondrial DNA structure, and then reduce intracellular ATP synthesis, eventually causing mitochondrial permeability change, and a large amount of oxidative active substances released, the latter directly attack cell biological macromolecules, causing cell oxidative damage. Oxidative stress and mitochondrial dysfunction play important roles in the pathogenesis of depression [9,10].

[1] Niki E , Yoshida Y , Saito Y , et al. Lipid peroxidation: Mechanisms, inhibition, and biological effects[J]. Biochemical & Biophysical Research Communications, 2005, 338(1):0-676.

[2] Sarandol A , Sarandol E , Eker S S , et al. Major depressive disorder is accompanied with oxidative stress: short-term antidepressant treatment does not alter oxidative-antioxidative systems [J]. Human Psychopharmacology: Clinical and Experimental, 2007, 22(2):67-73.

[3] Lipid peroxidation and antioxidant protection in patients during acute depressive episodes and in remission after fluoxetine treatment [J]. Pharmacological Reports, 2009, 61(3):436-447.

[4] Goshen I , Kreisel T , Ben-Menachem-Zidon O , et al. Brain interleukin-1 mediates chronic stress-induced depression in mice via adrenocortical activation and hippocampal neurogenesis suppression[J]. Molecular Psychiatry, 2008, 13(7):717-728.

[5] Olga J.G. Schiepers, Marieke C. Wichers Cytokines and major depression[J]. Progress in Neuro Psychopharmacology & Biological Psychiatry 2005,29(2):201–217.

[6] Black P H . Stress and the inflammatory response: A review of neurogenic inflammation [J]. Brain Behavior and Immunity, 2002, 16(6):0-653.

[7] Choi D K , Koppula S , Choi M , et al. Recent developments in the inhibitors of neuroinflammation and neurodegeneration: inflammatory oxidative enzymes as a drug target[J]. Expert Opinion on Therapeutic Patents, 2010, 20(11):1531-1546.

[8] Brocardo P S , Budni J , Kaster M P , et al. P.2.d.019 Antidepressant-like effect of the acute administration of folic acid in mice[J]. European Neuropsychopharmacology, 2006, 16(06):S343-S343.

[9] Rezin G T , Cardoso M R , Cinara L Gonçalves, et al. Inhibition of mitochondrial respiratory chain in brain of rats subjected to an experimental model of depression[J]. Neurochemistry International, 2008, 53(6-8):395-400.

[10] Gong Y , Chai Y , Ding J H , et al. Chronic mild stress damages mitochondrial ultrastructure and function in mouse brain[J]. Neuroscience Letters, 2010, 488(1):76-80.

- Only very few characteristics of the study population are presented and possible confounding cannot be assessed. Is information regarding inflammation markers of e.g. coronary heart disease available? Depression and coronary heart disease might be linked vi inflammation or one may increase the risk of developing the other.

Reply

The existing demographic characteristics were relatively small. Currently, only age, gender, and marriage included. But education, ethnicity, and ethnicity weren’t included. The reason for this is mainly related to the source of the control group selected in this study. The normal control group in this study was derived from the physical examination population in our hospital. (1) Researchers have not been able to obtain the education level information of medical examiners in the electronic information management system of the physical examination crowd. (2) The researchers included ethnic groups in the analysis of the data, but found that most of the ethnic groups studied were Han ethnic groups, and very few were ethnic minorities. Therefore, no special comparative analysis was performed on them. The information was ultimately not presented in the research results. (3) The patients diagnosed and treated by our medical institution are basically yellow races, and other races are basically non-existent, and the current patient information electronic record system of our hospital uses a text format when recording this information. For this reason, the demographic characteristics of race are not presented separately in the study results. During the design phase of this study, we considered the possible impact of the lack of demographic characteristics on the final results, so we consulted a statistical expert. He thought that our concerns were justified, but combined with the reasons we mentioned above, and considering that this study was a large sample study, it was believed that a small demographic feature will have an impact, but it should not be enough to change the final results of the study. After discussion by the research team, finally decided not to include the above demographic characteristics in the analysis. We will try our best to pay attention to the suggestions of reviewers when we do similar researches in the future.

The question of whether there are indicators of inflammatory markers for certain diseases such as coronary heart disease. We did not consider it during the study design and implementation stages, mainly because the uric acid involved in this study was not related to inflammatory markers of various diseases. However, during the data collection stage, the past history of patients was included, and some of them did have various chronic physical diseases.

- The authors first state that “all data are fully available without restriction” but then state that “Data cannot be shared publicly because of the request of our hospital”. So I think the first statement will have to be changed.

Reply

Amended and the data will be uploaded with the revised manuscript.

- There are some typos and repeatedly whitespaces are missing. The language should be revised. The unit of uric acid is given as umol//L instead of µmol/L.

Reply

Amended (see pages in the Revised Manuscript with Track Changes)

- In all the tables as well as in the text a number of p values is reported as 0.00. I would recommend to report three digits. Please report the actual p value and use something like p<0.001 for very low p values.

Reply

Amended (see pages in the Revised Manuscript with Track Changes)

- Table 1 is not referenced in the text.

Reply

Amended (see page5 in the Revised Manuscript with Track Changes)

- Why did the authors categorize age? They could use it as continuous variable.

Reply

Age was only used to analyze whether there was a statistical difference between the normal population and the patient population in the study (Table 1). It wasn’t included in the relevant statistical analysis of uric acid as a biological marker of depression. Generally speaking, to facilitate the description of age, it was categorized. At the same time, researchers have also referred to the description of age in the papers published by international journals. Finally, we decided to categorize age. 

- Was uric acid normally distributed? It would be interesting to look at the distribution in the different patient groups, perhaps in form of box plots.

Reply

The histogram and P-P chart were used to judge the distribution of the uric acid value. According to the result, the values followed the normal distribution.(Figure 1)

Figure 1 P-P diagram and histogram of uric acid distribution in patients

Reviewer #3: 

The manuscript described a technically sound piece of scientific research with data that supports the conclusions. and the experiments have been conducted rigorously, with appropriate controls, replication, and sample sizes. The conclusions have drawn appropriately based on the data presented. The manuscript is technically sound, and the data support the conclusions. But the authors have not made all data underlying the findings in their manuscript fully available. 

Reply

Amended. At the time of submission, we did not fully understand the requirements for data upload. After consulting our hospital's management agency, the data used in this study can be uploaded. We have uploaded the data while uploading the revised manuscript and noted this in the cover letter.

6. PLOS authors have the option to publish the peer review history of their article (what does this mean?). If published, this will include your full peer review and any attached files.

Do you want your identity to be public for this peer review? For information about this choice, including consent withdrawal, please see our Privacy Policy.

Reviewer #1: Yes: Shusuke Numata

Reviewer #2: No

Reviewer #3: No

Reply

I agree to publish the peer review history of their article, and want my identity to be public for this peer review.

Reply

There are no figures in this article.

---

## [Decision Letter · Decision Letter 1]

5 Feb 2020

PONE-D-19-30310R1

Serum Uric Acid a Depression Biomarker

PLOS ONE

Dear Mr. Meng,

Thank you for submitting your manuscript to PLOS ONE. After careful consideration, we feel that it has merit but does not fully meet PLOS ONE’s publication criteria as it currently stands. Therefore, we invite you to submit a revised version of the manuscript that addresses the points raised during the review process.

The reviewer #1 addressed minor concerns about your manuscript. Please revise the manuscript again.

We would appreciate receiving your revised manuscript by Mar 21 2020 11:59PM. To enhance the reproducibility of your results, we recommend that if applicable you deposit your laboratory protocols in protocols.io, where a protocol can be assigned its own identifier (DOI) such that it can be cited independently in the future. For instructions see: http://journals.plos.org/plosone/s/submission-guidelines#loc-laboratory-protocols

We look forward to receiving your revised manuscript.

Kind regards,

Kenji Hashimoto, PhD

Academic Editor

PLOS ONE

Reviewers' comments:

Reviewer's Responses to Questions

**Comments to the Author**

1. If the authors have adequately addressed your comments raised in a previous round of review and you feel that this manuscript is now acceptable for publication, you may indicate that here to bypass the “Comments to the Author” section, enter your conflict of interest statement in the “Confidential to Editor” section, and submit your "Accept" recommendation.

Reviewer #1: All comments have been addressed

Reviewer #2: All comments have been addressed

2. Is the manuscript technically sound, and do the data support the conclusions?

Reviewer #1: Partly

Reviewer #2: Yes

3. Has the statistical analysis been performed appropriately and rigorously? 

Reviewer #1: Yes

Reviewer #2: Yes

4. Have the authors made all data underlying the findings in their manuscript fully available?

Reviewer #1: Yes

Reviewer #2: Yes

5. Is the manuscript presented in an intelligible fashion and written in standard English?

Reviewer #1: Yes

Reviewer #2: Yes

6. Review Comments to the Author

Reviewer #1: I think that it is too strong to mention that SUA can be used as a biomarker for the differential diagnosis of some subtypes of depression and to determine the length of hospital stay. If the authors will not show specific efficacy of ureic acid as a biological marker in thier draft, the comments in their conclusion should be changed (ex. SUA levels of bipolar disorder depressive episode may be higher compared to major depressive disorder levels, and the length of hospital stay may be associated with serum uric acid levels.).

Reviewer #2: (No Response)

7. PLOS authors have the option to publish the peer review history of their article (what does this mean?). If published, this will include your full peer review and any attached files.

Reviewer #1: Yes: Numata Shusuke

Reviewer #2: No

---

## [Author Response · Author response to Decision Letter 1]

8 Feb 2020

Comments to the Author

1. If the authors have adequately addressed your comments raised in a previous round of review and you feel that this manuscript is now acceptable for publication, you may indicate that here to bypass the “Comments to the Author” section, enter your conflict of interest statement in the “Confidential to Editor” section, and submit your "Accept" recommendation.

Reviewer #1: All comments have been addressed

Reviewer #2: All comments have been addressed

2. Is the manuscript technically sound, and do the data support the conclusions?

Reviewer #1: Partly

Reply

Amended. The reviewer pointed that the conclusion that serum uric acid was a biological marker of depression was somewhat inadequate based on the results of the current study. In this regard, after discussion by the research team, we adopted the recommendations of the reviewer and partially modified the research conclusions. (page1 and page9)

Reviewer #2: Yes

3. Has the statistical analysis been performed appropriately and rigorously? 

Reviewer #1: Yes

Reviewer #2: Yes

4. Have the authors made all data underlying the findings in their manuscript fully available?

Reviewer #1: Yes

Reviewer #2: Yes

5. Is the manuscript presented in an intelligible fashion and written in standard English?

Reviewer #1: Yes

Reviewer #2: Yes

6. Review Comments to the Author

Reviewer #1: I think that it is too strong to mention that SUA can be used as a biomarker for the differential diagnosis of some subtypes of depression and to determine the length of hospital stay. If the authors will not show specific efficacy of ureic acid as a biological marker in thier draft, the comments in their conclusion should be changed (ex. SUA levels of bipolar disorder depressive episode may be higher compared to major depressive disorder levels, and the length of hospital stay may be associated with serum uric acid levels.).

Reply

Amended. After discussion by the research team, we adopted the recommendations of the reviewer and partially modified the research conclusions. (Page1,Page8,and Page9)

Reviewer #2: (No Response)

7. PLOS authors have the option to publish the peer review history of their article (what does this mean?). If published, this will include your full peer review and any attached files.

Do you want your identity to be public for this peer review? For information about this choice, including consent withdrawal, please see our Privacy Policy.

Reviewer #1: Yes: Numata Shusuke

Reviewer #2: No

Reply

The researcher logged on the website you mentioned above, and entered the relevant interface after registration, but found that the website only edits figures, and the results of this study were presented in tables, so the research team believes that we needn’t do this. We are not quite sure whether our understanding is correct. If we understand incorrectly, we hope you give us the opportunity to make changes.

---

## [Editor Report · Decision Letter 2]

11 Feb 2020

Serum Uric Acid a Depression Biomarker

PONE-D-19-30310R2

Dear Dr. Meng,

We are pleased to inform you that your manuscript has been judged scientifically suitable for publication and will be formally accepted for publication once it complies with all outstanding technical requirements.

With kind regards,

Kenji Hashimoto, PhD

Section Editor

PLOS ONE
---

## [Editor Report · Acceptance letter]

19 Feb 2020

PONE-D-19-30310R2 

Serum Uric Acid a Depression Biomarker 

Dear Dr. Meng:

I am pleased to inform you that your manuscript has been deemed suitable for publication in PLOS ONE. Congratulations! Your manuscript is now with our production department. 

With kind regards,

on behalf of

Prof. Kenji Hashimoto 

Section Editor

PLOS ONE